# BUILDING EFFECTIVE DEEP NEURAL NETWORKS ONE FEATURE AT A TIME

## ABSTRACT

Successful training of convolutional neural networks is often associated with sufficiently deep architectures composed of high amounts of features. These networks typically rely on a variety of regularization and pruning techniques to converge to less redundant states. We introduce a novel bottom-up approach to expand representations in fixed-depth architectures. These architectures start from just a single feature per layer and greedily increase width of individual layers to attain effective representational capacities needed for a specific task. While network growth can rely on a family of metrics, we propose a computationally efficient version based on feature time evolution and demonstrate its potency in determining feature importance and a networks' effective capacity. We demonstrate how automatically expanded architectures converge to similar topologies that benefit from lesser amount of parameters or improved accuracy and exhibit systematic correspondence in representational complexity with the specified task. In contrast to conventional design patterns with a typical monotonic increase in the amount of features with increased depth, we observe that CNNs perform better when there is more learnable parameters in intermediate, with falloffs to earlier and later layers.

## 1 INTRODUCTION

Estimating and consequently adequately setting representational capacity in deep neural networks for any given task has been a long standing challenge. Fundamental understanding still seems to be insufficient to rapidly decide on suitable network sizes and architecture topologies. While widely adopted convolutional neural networks (CNNs) such as proposed by Krizhevsky et al. (2012); Simonyan & Zisserman (2015); He et al. (2016); Zagoruyko & Komodakis (2016) demonstrate high accuracies on a variety of problems, the memory footprint and computational complexity vary. An increasing amount of recent work is already providing valuable insights and proposing new methodology to address these points. For instance, the authors of Baker et al. (2016) propose a reinforcement learning based meta-learning approach to have an agent select potential CNN layers in a greedy, yet iterative fashion. Other suggested architecture selection algorithms draw their inspiration from evolutionary synthesis concepts (Shafiee et al., 2016; Real et al., 2017). Although the former methods are capable of evolving architectures that rival those crafted by human design, it is currently only achievable at the cost of navigating large search spaces and hence excessive computation and time. As a trade-off in present deep neural network design processes it thus seems plausible to consider layer types or depth of a network to be selected by an experienced engineer based on prior knowledge and former research. A variety of techniques therefore focus on improving already well established architectures. Procedures ranging from distillation of one network's knowledge into another (Hinton et al., 2014), compressing and encoding learned representations Han et al. (2016), pruning alongside potential re-training of networks (Han et al., 2015; 2017; Shrikumar et al., 2016; Hao et al., 2017), small capacity increases on trained networks in transfer-learning scenarios (Warde-Farley et al., 2014) and the employment of different regularization terms during training (He et al., 2015; Kang et al., 2016; Rodriguez et al., 2017; Alvarez & Salzmann, 2016), are just a fraction of recent efforts in pursuit of reducing representational complexity while attempting to retain accuracy. Underlying mechanisms rely on a multitude of criteria such as activation magnitudes (Shrikumar et al., 2016) and small weight values (Han et al., 2015) that are used as pruning metrics for either single neurons or complete feature maps, in addition to further combination with regularization and penalty terms.

Common to these approaches is the necessity of training networks with large parameter quantities for maximum representational capacity to full convergence and the lack of early identification of insufficient capacity. In contrast, this work proposes a bottom-up approach with the following contributions:

- We introduce a computationally efficient, intuitive metric to evaluate feature importance at any point of training a neural network. The measure is based on feature time evolution, specifically the normalized cross-correlation of each feature with its initialization state.

- We propose a bottom-up greedy algorithm to automatically expand fixed-depth networks that start with one feature per layer until adequate representational capacity is reached. We base addition of features on our newly introduced metric due to its computationally efficient nature, while in principle a family of similarly constructed metrics is imaginable.

- We revisit popular CNN architectures and compare them to automatically expanded networks. We show how our architectures systematically scale in terms of complexity of different datasets and either maintain their reference accuracy at reduced amount of parameters or achieve better results through increased network capacity.

- We provide insights on how evolved network topologies differ from their reference counterparts where conventional design commonly increases the amount of features monotonically with increasing network depth. We observe that expanded architectures exhibit increased feature counts at early to intermediate layers and then proceed to decrease in complexity.

## 2 BUILDING NEURAL NETWORKS BOTTOM-UP FEATURE BY FEATURE

While the choice and size of deep neural network model indicate the *representational capacity* and thus determine which functions can be learned to improve training accuracy, training of neural networks is further complicated by the complex interplay of choice of optimization algorithm and model regularization. Together, these factors define define the *effective capacity*. This makes training of deep neural networks particularly challenging. One practical way of addressing this challenge is to boost model sizes at the cost of increased memory and computation times and then applying strong regularization to avoid over-fitting and minimize generalization error. However, this approach seems unnecessarily cumbersome and relies on the assumption that optimization difficulties are not encountered. We draw inspiration from this challenge and propose a bottom-up approach to increase capacity in neural networks along with a new metric to gauge the effective capacity in the training of (deep) neural networks with stochastic gradient descent (SGD) algorithms.

### 2.1 NORMALIZED WEIGHT-TENSOR CROSS-CORRELATION AS A MEASURE FOR NEURAL NETWORK EFFECTIVE CAPACITY

In SGD the objective function $J(\Theta)$ is commonly equipped with a penalty on the parameters $R(\Theta)$, yielding a regularized objective function:

$$\hat{J}(\Theta) = J(\Theta) + \alpha R(\Theta) . \tag{1}$$

Here, $\alpha$ weights the contribution of the penalty. The regularization term $R(\Theta)$ is typically chosen as a $L_2$-norm, coined weight-decay, to decrease model capacity or a $L_1$-norm to enforce sparsity. Methods like dropout (Srivastava et al., 2014) and batch normalization (Ioffe & Szegedy, 2015) are typically employed as further implicit regularizers.

In principle, our approach is inspired by earlier works of Hao et al. (2017) who measure a complete feature's importance by taking the $L_1$-norm of the corresponding weight tensor instead of operating on individual weight values. In the same spirit we assign a single importance value to each feature based on its values. However we do not use the weight magnitude directly and instead base our metric on the following hypothesis: While a feature's absolute magnitude or relative change between two subsequent points in time might not be adequate measures for direct importance, the relative amount of change a feature experiences with respect to its original state provides an indicator for how many times and how much a feature is changed when presented with data. Intuitively we suggest that features that experience high structural changes must play a more vital role than any feature that is initialized and does not deviate from its original states' structure. There are two potential reasons why a feature that has randomly been initialized does not change in structure: The

first being that its form is already initialized so well that it does not need to be altered and can serve either as is or after some scalar rescaling or shift in order to contribute. The second possibility is that too high representational capacity, the nature of the cost function, too large regularization or the type of optimization algorithm prohibit the feature from being learned, ultimately rendering it obsolete. As deep neural networks are commonly initialized from using a distribution over high-dimensional space the first possibility seems unlikely (Goodfellow et al., 2016).

As one way of measuring the effective capacity at a given state of learning, we propose to monitor the time evolution of the normalized cross-correlation for all weights with respect to their state at initialization. For a neural network composed of layers $l = 1, 2, \ldots, L - 1$ and complementary weight-tensors $\mathbf{W}^l_{f^l f^{l+1}}$, with optional spatial dimensions $j^l \times k^l$ (e.g. for convolutional filters), defining a mapping from an input feature-space $f^l = 1, 2, \ldots F^l$ onto the output feature space $f^{l+1} = 1, 2, \ldots F^{l+1}$ that serves as input to the next layer, we define the following metric:

$$\mathbf{c}^l_{f^{l+1},t} = 1 - \frac{\sum_{f^l} \left[ \left( \mathbf{W}^l_{f^l f^{l+1}, t_0} - \bar{\mathbf{W}}^l_{f^{l+1}, t_0} \right) \circ \left( \mathbf{W}^l_{f^l f^{l+1}, t} - \bar{\mathbf{W}}^l_{f^{l+1}, t} \right) \right]}{\left\| \mathbf{W}^l_{t_0} \right\|_{2, f^{l+1}} \cdot \left\| \mathbf{W}^l_t \right\|_{2, f^{l+1}}} \tag{2}$$

which is a measure of self-resemblance based on the normalized cross-correlation of a filter's state at two different points in time. In this equation, $\mathbf{W}^l_{f^l f^{l+1}, t}$ is the state of a layer's weight-tensor at time $t$ or the initial state after initialization $t_0$. $\bar{\mathbf{W}}^l_{f^{l+1}, t}$ is the mean taken over input feature and potential spatial dimensions. $\circ$ depicts the Hadamard product that we use in an extended fashion from matrices to tensors where each dimension is multiplied in an element-wise fashion analogously. Similarly the terms in the denominator are defined as the $L_2$-norm of the weight-tensor taken over said dimensions and thus resulting in a scalar value. Above equation is applicable to multi-layer perceptrons as well as features with spatial dimensionality, where the sum over the input feature space $F^l$ then also includes a feature's spatial dimensions $j^l \times k^l$.

The metric is easily interpretable as no structural changes of features lead to a value of zero and importance approaches unity the more a feature is deviating in structure. The usage of normalized cross-correlation with the $L_2$-norm in the denominator has the advantage of having an inherent invariance to effects such as translations or rescaling of weights stemming from various regularization contributions. Therefore the contribution of the sum-term in equation 1 does not change the value of the metric if the gradient term vanishes. This is in contrast to the measure proposed by Hao et al. (2017), as absolute weight magnitudes are affected by rescaling and make it more difficult to interpret the metric in an absolute way and find corresponding thresholds. Due to this normalized nature of our metric we are able to move away from a top-down pruning approach, as presented by Hao et al. (2017), altogether and instead follow a bottom-up procedure where we incrementally add features, eradicating the need to train a large architecture in the first place.

## 2.2 BOTTOM-UP CONSTRUCTION OF NEURAL NETWORK REPRESENTATIONAL CAPACITY

We propose a new method to converge to architectures that encapsulate necessary task complexity without the necessity of training huge networks in the first place. Starting with one feature in each layer, we expand our architecture as long as the effective capacity, as estimated through our metric, is not met and all features experience structural change. In contrast to methods such as Baker et al. (2016); Shafiee et al. (2016); Real et al. (2017) we do not consider flexible depth and treat the amount of layers in a network as a prior based on the belief of hierarchical composition of the underlying factors. This fixed-depth prior is similar to the approach of Philipp & Carbonell (2017), who modify fixed-depth fully-connected networks, albeit without explicitly introducing capacity as a term in the optimization loss, but as a modular auxiliary step on top of a conventional SGD algorithm. Our method, shown in algorithm 1, can be summarized as follows:

1. For a given network arrangement in terms of function type, depth and a set of hyper-parameters: initialize each layer with one feature and proceed with (mini-batch) SGD.

2. After each update step evaluate equation 2 independently per layer and increase feature dimensionality by $F_{exp}$ (one or higher if a complexity prior exists) if all currently present features in respective layer are differing from their initial state by more than a constant $\epsilon$.

3. Re-initialize all parameters if architecture has expanded.

---

**Algorithm 1** Greedy architecture feature expansion algorithm

---

**Require:** Set hyper-parameters: learning rate $\lambda_0$, mini-batch size, maximum epoch $t_{end}$, ...
**Require:** Set expansion parameters: $\epsilon = 10^{-6}$, $F_{exp} = 1$ (or higher)
1: Initialize parameters: $t = 1$, $F^l = 1 \, \forall l = 1, 2, \ldots L - 1$, $\Theta$ , $reset = false$
2: **while** $t \le t_{end}$ **do**
3:      **for** mini-batches in training set **do**
4:         $reset \leftarrow false$
5:         Compute gradient and perform update step
6:         **for** $l = 1$ to $L - 1$ **do**
7:            **for** $i = f^{l+1}$ to $F^{l+1}$ **do**    ⎫
8:              Update $\mathbf{c}^l_{i,t}$ according to equation 2   ⎬  in parallel.
9:            **end for**    ⎭
10:           **if** $\max(\mathbf{c}^l_t) < 1 - \epsilon$ **then**
11:             $F^{l+1} \leftarrow F^{l+1} + F_{exp}$
12:             $reset \leftarrow true$
13:           **end if**
14:         **end for**
15:         **if** $reset == true$ **then**
16:           Re-initialize parameters $\Theta$, $t = 0$, $\lambda = \lambda_0$, ...
17:         **end if**
18:      **end for**
19:      $t \leftarrow t + 1$
20: **end while**

---

The constant $\epsilon$ is a numerical stability parameter that we set to a small value such as $10^{-6}$, but could in principle as well be used as a constraint. We have decided to include the re-initialization in step 3 (lines $15 - 17$) to avoid the pitfalls of falling into local minima (see appendix section A.5 for a brief discussion on the need of re-initialization). Despite this sounding like a major detriment to our method, we show that networks nevertheless rapidly converge to a stable architectural solution that comes at less than perchance expected computational overhead and at the benefit of avoiding training of too large architectures. Naturally at least one form of explicit or implicit regularization has to be present in the learning process in order to prevent infinite expansion of the architecture. We would like to emphasize that we have chosen the metric defined in equation 2 as a basis for the decision of when to expand an architecture, but in principle a family of similarly constructed metrics is imaginable. We have chosen this particular metric because it does not directly depend on gradient or higher-order term calculation and only requires multiplication of weights with themselves. The algorithm furthermore doesn't intervene in the SGD optimization step as the formulation doesn't involve alteration of the cost function. Thus, a major advantage is that computation of equation 2 can be executed modularly and independently on top of a conventional DNN optimization procedure, can be parallelized completely and therefore executed at less cost than a regular forward pass through the network.

## 3   Revisiting popular architectures with architecture expansion

We revisit some of the most established architectures "GFCNN" (Goodfellow et al., 2013) "VGG-A & E" (Simonyan & Zisserman, 2015) and "Wide Residual Network: WRN" (Zagoruyko & Komodakis, 2016) (see appendix for architectural details) with batch normalization (Ioffe & Szegedy, 2015). We compare the number of learnable parameters and achieved accuracies with those obtained through expanded architectures that started from a single feature in each layer. For each architecture we include all-convolutional variants (Springenberg et al., 2015) that are similar to WRNs (minus the skip-connections), where all pooling layers are replaced by convolutions with larger stride. All fully-connected layers are furthermore replaced by a single convolution (affine, no activation function) that maps directly onto the space of classes. Even though the value of more complex type of sub-sampling functions has already empirically been demonstrated (Lee et al., 2015), the amount of features of the replaced layers has been constrained to match in dimensionality with the preceding convolution layer. We would thus like to further extend and analyze the role of layers involving

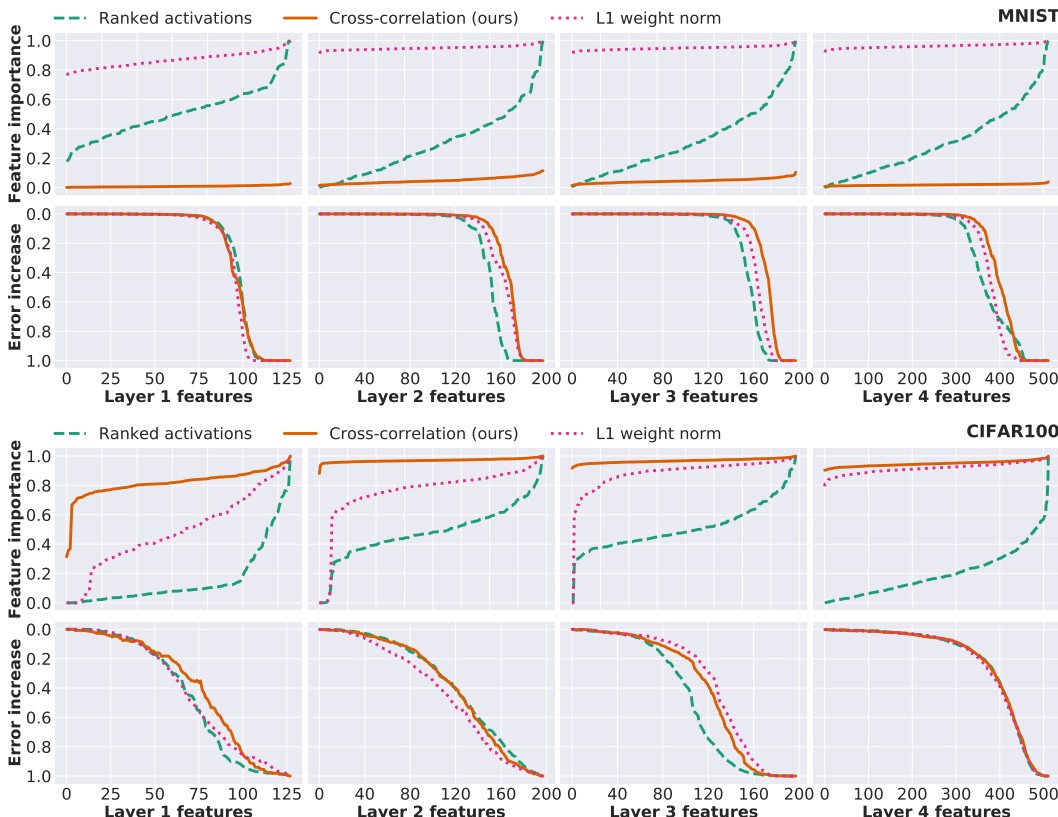

Figure 1: Pruning of complete features for the GFCNN architecture trained on MNIST (top panel) and CIFAR100 (bottom panel). Top row shows sorted feature importance values for every layer according to three different metrics at the end of training. Bottom row illustrates accuracy loss when removing feature by feature in ascending order of feature importance.

sub-sampling by decoupling the dimensionality of these larger stride convolutional layers.

We consider these architectures as some of the best CNN architectures as each of them has been chosen and tuned carefully according to extensive amounts of hyper-parameter search. As we would like to demonstrate how representational capacity in our automatically constructed networks scales with increasing task difficulty, we perform experiments on the MNIST (LeCun et al., 1998), CIFAR10 & 100 (Krizhevsky, 2009) datasets that intuitively represent little to high classification challenge. We also show some preliminary experiments on the ImageNet (Russakovsky et al., 2015) dataset with "Alexnet" (Krizhevsky et al., 2012) to conceptually show that the algorithm is applicable to large scale challenges. All training is closely inspired by the procedure specified in Zagoruyko & Komodakis (2016) with the main difference of avoiding heavy preprocessing. We preprocess all data using only trainset mean and standard deviation (see appendix for exact training parameters). Although we are in principle able to achieve higher results with different sets of hyper-parameters and preprocessing methods, we limit ourselves to this training methodology to provide a comprehensive comparison and avoid masking of our contribution. We train all architectures five times on each dataset using a Intel i7-6800K CPU (data loading) and a single NVIDIA Titan-X GPU. Code has been written in both Torch7 (Collobert et al., 2011) and PyTorch (http://pytorch.org/) and will be made publicly available.

## 3.1 THE TOP-DOWN PERSPECTIVE: FEATURE IMPORTANCE FOR PRUNING

We first provide a brief example for the use of equation 2 through the lens of pruning to demonstrate that our metric adequately measures feature importance. We evaluate the contribution of the features by pruning the weight-tensor feature by feature in ascending order of feature importance values and re-evaluating the remaining architecture. We compare our normalized cross-correlation metric 2 to

the $L_1$ weight norm metric introduced by Hao et al. (2017) and ranked mean activations evaluated over an entire epoch. In figure 1 we show the pruning of a trained GFCNN, expecting that such a network will be too large for the easier MNIST and too small for the difficult CIFAR100 task. For all three metrics pruning any feature from the architecture trained on CIFAR100 immediately results in loss of accuracy, whereas the architecture trained on MNIST can be pruned to a smaller set of parameters by greedily dropping the next feature with the currently lowest feature importance value. We notice how all three metrics perform comparably. However, in contrast to the other two metrics, our normalized cross-correlation captures whether a feature is important on absolute scale. For MNIST the curve is very close to zero, whereas the metric is close to unity for all CIFAR100 features. Ultimately this is the reason our metric, in the way formulated in equation 2, is used for the algorithm presented in 1 as it doesn't require a difficult process to determine individual layer threshold values. Nevertheless it is imaginable that similar metrics based on other tied quantities (gradients, activations) can be formulated in analogous fashion.

As our main contribution lies in the bottom-up widening of architectures we do not go into more detailed analysis and comparison of pruning strategies. We also remark that in contrast to a bottom-up approach to finding suitable architectures, pruning seems less desirable. It requires convergent training of huge architectures with lots of regularization before complexity can be reduced, pruning is not capable of adding complexity if representational capacity is lacking, pruning percentages are difficult to interpret and compare (i.e. pruning percentage is 0 if the architecture is adequate), a majority of parameters are pruned only in the last "fully-connected" layers (Han et al., 2015), and pruning strategies as suggested by Han et al. (2015; 2017); Shrikumar et al. (2016); Hao et al. (2017) tend to require many cross-validation with consecutive fine-tuning steps. We thus continue with the bottom-up perspective of expanding architectures from low to high representational capacity.

### 3.2 THE BOTTOM-UP PERSPECTIVE: EXPANDING ARCHITECTURES

We use the described training procedure in conjunction with algorithm 1 to expand representational complexity by adding features to architectures that started with just one feature per layer with the following additional settings:

**Architecture expansion settings and considerations:** Our initial experiments added one feature at a time, but large speed-ups can be introduced by means of adding stacks of features. Initially, we avoided suppression of late re-initialization to analyze the possibility that rarely encountered worst-case behavior of restarting on an almost completely trained architecture provides any benefit. After some experimentation our final report used a stability parameter ending the network expansion if half of the training has been stable (no further change in architecture) and added $F_{exp} = 8$ and $F_{exp} = 16$ features per expansion step for MNIST and CIFAR10 & 100 experiments respectively.

We show an exemplary architecture expansion of the GFCNN architecture's layers for MNIST and CIFAR100 datasets in figure 2 and the evolution of the overall amount of parameters for five different experiments. We observe that layers expand independently at different points in time and more features are allocated for CIFAR100 than for MNIST. When comparing the five different runs we can identify that all architectures converge to a similar amount of network parameters, however at different points in time. A good example to see this behavior is the solid (green) curve in the MNIST example, where the architecture at first seems to converge to a state with lower amount of parameters and after some epochs of stability starts to expand (and re-initialize) again until it ultimately converges similarly to the other experiments.

We continue to report results obtained for the different datasets and architectures in table 1. The table illustrates the mean and standard deviation values for error, total amount of parameters and the mean overall time taken for five runs of algorithm 1 (tentative as heavily dependent on $F_{exp}$. Deviations can be fairly large due to the behavior observed in 2). We make the following observations:

- Without any prior on layer widths, expanding architectures converge to states with at least similar accuracies to the reference at reduced amount of parameters, or better accuracies by allocating more representational capacity.

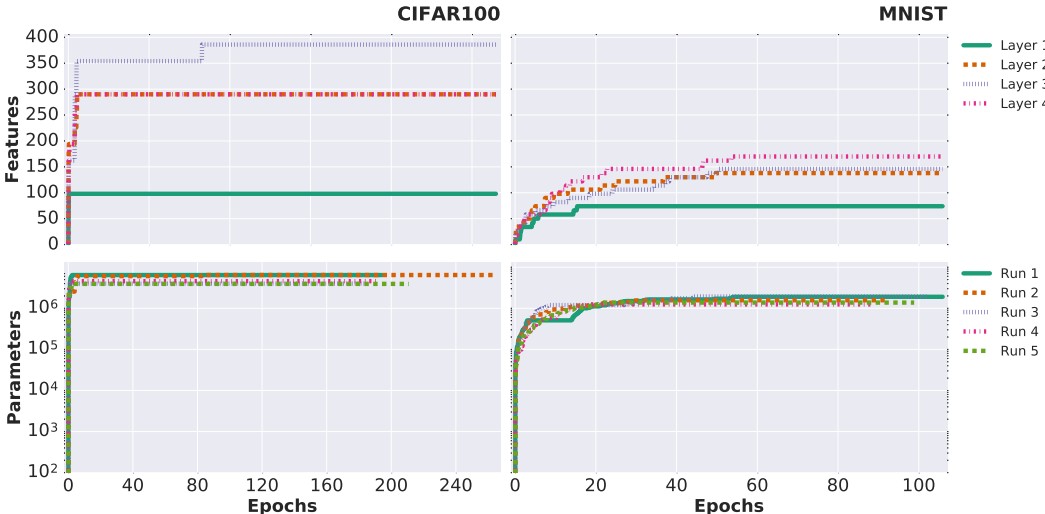

Figure 2: Exemplary GFCNN network expansion on MNIST and CIFAR100. Top panel shows one architecture's individual layer expansion; bottom panel shows the evolution of total parameters for five runs. It is observable how different experiments converge to similar network capacity on slightly different time-scales and how network capacity systematically varies with complexity of the dataset.

Table 1: Mean error and standard deviation and number of parameters (in millions) for architectures trained five times using the original reference and automatically expanded architectures respectively. For MNIST no data augmentation is applied. We use minor augmentation (flips, translations) for CIFAR10 & 100. All-convolutional (all-conv) versions have been evaluated for each architecture (except WRN where convolutions are stacked already). The * indicates hardware limitation.

| | | | GFCNN | | VGG-A | | VGG-E | | WRN-28-10 | |
|---|---|---|---|---|---|---|---|---|---|---|
| | type | | original | expanded | original | expanded | original | expanded | original | expanded |
| MNIST | standard | error [%] | 0.487 | 0.528 ±0.03 | 0.359 | 0.394 ±0.05 | 0.386 | 0.388 ±0.03 | overfit | 0.392±0.05 |
| | | params [M] | 4.26 | 1.61 ±0.31 | 13.70 | 3.35 ±0.45 | 20.57 | 6.49 ±0.89 | 36.48 | 4.83 ±0.57 |
| | | time [h] | 0.29 | 0.90 ±0.32 | 0.59 | 5.35 ±1.95 | 0.48 | 9.8 ±3.02 | 2.47 | 2.91 ±0.28 |
| | all-conv | error [%] | 0.535 | 0.552 ±0.03 | 0.510 | 0.502 ±0.04 | 0.523 | 0.528 ±0.04 | | |
| | | params [M] | 3.71 | 2.19 ±0.55 | 10.69 | 3.79 ±0.57 | 21.46 | 6.02 ±0.73 | | |
| | | time [h] | 0.39 | 1.68 ±1.05 | 0.64 | 3.06 ±0.97 | 0.52 | 6.58 ±2.81 | | |
| CIFAR10+ | standard | error [%] | 11.32 | 11.03 ±0.19 | 7.18 | 6.73 ±0.05 | 7.51 | 5.64 ±0.11 | 4.04 | 3.95 ±0.12 |
| | | params [M] | 4.26 | 4.01 ±0.62 | 13.70 | 8.54 ±1.51 | 20.57 | 27.41 ±4.09 | 36.48 | 25.30 ±1.62 |
| | | time [h] | 0.81 | 1.40 ±0.22 | 1.61 | 3.95 ±0.59 | 1.32 | 16.32 ±5.39 | 8.22 | 21.18 ±2.12 |
| | all-conv | error [%] | 8.78 | 8.13 ±0.11 | 6.71 | 6.56 ±0.18 | 7.46 | 5.42 ±0.11 | | |
| | | params | 3.71 | 10.62 ±1.91 | 10.69 | 8.05 ±1.57 | 21.46 | 44.98 ±7.31 | | |
| | | time [h] | 1.19 | 3.38 ±0.76 | 1.74 | 5.24 ±1.11 | 1.54 | 26.46 ±9.77 | | |
| CIFAR100+ | standard | error [%] | 34.91 | 34.23 ±0.29 | 25.01 | 25.17 ±0.34 | 29.43 | 25.06 ±0.55 | 18.51 | 18.44* |
| | | params [M] | 4.26 | 6.82 ±1.08 | 13.70 | 8.48 ±1.40 | 20.57 | 28.41 ±2.26 | 36.48 | 27.75* |
| | | time [h] | 0.81 | 1.83 ±0.56 | 1.61 | 3.83 ±0.47 | 1.32 | 16.67 ±2.89 | 8.22 | 13.9* |
| | all-conv | error [%] | 29.83 | 28.34 ±0.43 | 24.30 | 23.95 ±0.28 | 31.94 | 24.87 ±0.16 | | |
| | | params [M] | 3.71 | 21.40 ±3.71 | 10.69 | 10.84 ±2.41 | 21.46 | 44.59 ±4.49 | | |
| | | time [h] | 1.19 | 4.72 ±1.15 | 1.74 | 5.38 ±1.46 | 1.54 | 22.76 ±3.94 | | |

- For each architecture type there is a clear trend in network capacity that is increasing with dataset complexity from MNIST to CIFAR10 to CIFAR100 [1].

- Even though we have introduced re-initialization of the architecture the time taken by algorithm 1 is much less than one would invest when doing a manual, grid- or random-search.

---

[1] For the WRN CIFAR100 architecture the * signifies hardware memory limitations due to the arrangement of architecture topology and thus expansion is limited. This is because increased amount of early-layer features requires more memory in contrast to late layers, which is particularly intense for the coupled WRN architecture.

- Shallow GFCNN architectures are able to gain accuracy by increasing layer width, although there seems to be a natural limit to what width alone can do, especially when heavy regularization is in place (we discuss such a non-trivial example with corresponding curves in appendix section A.4). This is in agreement with observations pointed out in other works such as Ba & Caurana (2014); Urban et al. (2017).

- The large reference VGG-E (lower accuracy than VGG-A on CIFAR) and WRN-28-10 (complete over-fit on MNIST) seem to run into optimization difficulties for these datasets. However, expanded alternate architecture clearly perform significantly better.

In general we observe that these benefits are due to unconventional, yet always coinciding, network topology of our expanded architectures. These topologies suggest that there is more to CNNs than simply following the rule of thumb of increasing the number of features with increasing architectural depth. Before proceeding with more detail on these alternate architecture topologies, we want to again emphasize that we do not report experiments containing extended methodology such as excessive preprocessing, data augmentation, the oscillating learning rates proposed in Loshchilov & Hutter (2017) or better sets of hyper-parameters for reasons of clarity, even though accuracies rivaling state-of-the-art performances can be achieved in this way.

### 3.3 ALTERNATE FORMATION OF DEEP NEURAL NETWORK TOPOLOGIES

Almost all popular convolutional neural network architectures follow a design pattern of monotonically increasing feature amount with increasing network depth (LeCun et al., 1998; Goodfellow et al., 2013; Simonyan & Zisserman, 2015; Springenberg et al., 2015; He et al., 2016; Zagoruyko & Komodakis, 2016; Loshchilov & Hutter, 2017; Urban et al., 2017). For the results presented in table 1 all automatically expanded network topologies present alternatives to this pattern. In figure 3, we illustrate exemplary mean topologies for a VGG-E and VGG-E all-convolutional network

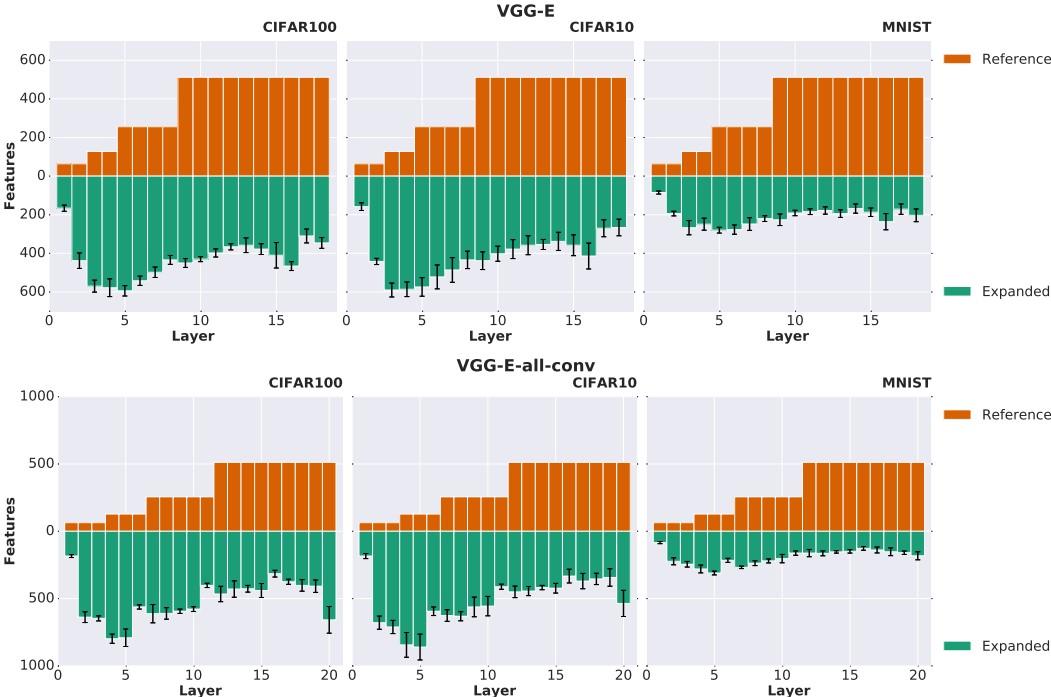

Figure 3: Mean and standard deviation of topologies as evolved from the expansion algorithm for a VGG-E and VGG-E all-convolutional architecture run five times on MNIST, CIFAR10 and CIFAR100 datasets respectively. Top panels show the reference architecture, whereas bottom shows automatically expanded architecture alternatives. Expanded architectures vary in capacity with dataset complexity and topologically differ from their reference counterparts.

as constructed by our expansion algorithm in five runs on the three datasets. Apart from noticing the systematic variations in representational capacity with dataset difficulty, we furthermore find topological convergence with small deviations from one training to another. We observe the highest feature dimensionality in early to intermediate layers with generally decreasing dimensionality towards the end of the network differing from conventional CNN design patterns. Even if the expanded architectures sometimes do not deviate much from the reference parameter count, accuracy seems to be improved through this topological re-arrangement. For architectures where pooling has been replaced with larger stride convolutions we also observe that dimensionality of layers with sub-sampling changes independently of the prior and following convolutional layers suggesting that highly-complex sub-sampling operations are learned. This an extension to the proposed all-convolutional variant of Springenberg et al. (2015), where introduced additional convolutional layers were constrained to match the dimensionality of the previously present pooling operations.

If we view the deep neural network as being able to represent any function that is limited rather by concepts of continuity and boundedness instead of a specific form of parameters, we can view the minimization of the cost function as learning a functional mapping instead of merely adopting a set of parameters (Goodfellow et al., 2016). We hypothesize that evolved network topologies containing higher feature amount in early to intermediate layers generally follow a process of first mapping into higher dimensional space to effectively separate the data into many clusters. The network can then more readily aggregate specific sets of features to form clusters distinguishing the class subsets. Empirically this behavior finds confirmation in all our evolved network topologies that are visualized in the appendix. Similar formation of topologies, restricted by the dimensionality constraint of the identity mappings, can be found in the trained residual networks.

While He et al. (2015) has shown that deep VGG-like architectures do not perform well, an interesting question for future research could be whether plainly stacked architectures can perform similarly to residual networks if the arrangement of feature dimensionality is differing from the conventional design of monotonic increase with depth.

## 3.4 An outlook to ImageNet

We show two first experiments on the ImageNet dataset using an all-convolutional Alexnet to show that our methodology can readily be applied to large scale. The results for the two runs can be found in table 2 and corresponding expanded architectures are visualized in the appendix. We observe that the experiments seem to follow the general pattern and again observe that topological rearrangement of the architecture yields substantial benefits. In the future we would like to extend experimentation to more promising ImageNet architectures such as deep VGG and residual networks. However, these architectures already require 4-8 GPUs and large amounts of time in their baseline evaluation, which is why we presently are not capable of evaluating these architectures and keep this section at a very brief proof of concept level.

Table 2: Two experiments with all-convolutional Alexnet on the large scale Imagenet dataset comparing the reference implementation with our expanded architecture.

| | Alexnet - 1 | | | | Alexnet - 2 | | | |
|---|---|---|---|---|---|---|---|---|
| | top-1 error | top-5 error | params | time | top-1 error | top-5 error | params | time |
| original | 43.73 % | 20.11 % | 35.24 M | 27.99 h | 43.73 % | 20.11 % | 35.24 M | 27.99 h |
| expanded | 37.84 % | 15.88 % | 34.76 M | 134.21 h | 38.47 % | 16.33 % | 32.98 M | 118.73 h |

## 4 Conclusion

In this work we have introduced a novel bottom-up algorithm to start neural network architectures with one feature per layer and widen them until a task depending suitable representational capacity is achieved. For the use in this framework we have presented one potential computationally efficient and intuitive metric to gauge feature importance. The proposed algorithm is capable of expanding architectures that provide either reduced amount of parameters or improved accuracies through higher amount of representations. This advantage seems to be gained through alternative network topologies with respect to commonly applied designs in current literature. Instead of increasing the amount of features monotonically with increasing depth of the network, we empirically observe that

expanded neural network topologies have high amount of representations in early to intermediate layers.

Future work could include a re-evaluation of plainly stacked deep architectures with new insights on network topologies and extended evaluation on different domain data. We have furthermore started to replace the currently present re-initialization step in the proposed expansion algorithm by keeping learned filters. In principle this approach looks promising but does need further systematic analysis of new feature initialization with respect to the already learned feature subset and accompanied investigation of orthogonality to avoid falling into local minima.

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

# A  APPENDIX

## A.1  DATASETS

- MNIST (LeCun et al., 1998): 50000 train images of hand-drawn digits of spatial size $28 \times 28$ belonging to one of 10 equally sampled classes.

- CIFAR10 & 100 (Krizhevsky, 2009): 50000 natural train images of spatial size $32 \times 32$ each containing one object belonging to one of 10/100 equally sampled classes.

- ImageNet (Russakovsky et al., 2015): Approximately 1.2 million training images of objects belonging to one of 1000 classes. Classes are not equally sampled with 732-1300 images per class. Dataset contains 50 000 validation images, 50 per class. Scale of objects and size of images varies.

## A.2  TRAINING HYPER-PARAMETERS

All training is closely inspired by the procedure specified in Zagoruyko & Komodakis (2016) with the main difference of avoiding heavy preprocessing. Independent of dataset, we preprocess all data using only trainset mean and standard deviation. All training has been conducted using cross-entropy as a loss function and weight initialization following the normal distribution as proposed by He et al. (2015). All architectures are trained with batch-normalization with a constant of $1 \cdot 10^{-3}$, a batch-size of 128, a $L_2$ weight-decay of $5 \cdot 10^{-4}$, a momentum of 0.9 and nesterov momentum.

**Small datasets:**  We use initial learning rates of 0.1 and 0.005 for the CIFAR and MNIST datasets respectively. We have rescaled MNIST images to $32 \times 32$ (CIFAR size) and repeat the image across color channels in order to use architectures without modifications. CIFAR10 & 100 are trained for 200 epochs and the learning rate is scheduled to be reduced by a factor of 5 every multiple of 60 epochs. MNIST is trained for 60 epochs and learning rate is reduced by factor of 5 once after 30 epochs. We augment the CIFAR10 & 100 training by introducing horizontal flips and small translations of up to 4 pixels during training. No data augmentation has been applied to the MNIST dataset.

**ImageNet:**  We use the single-crop technique where we rescale the image such that the shorter side is equal to 224 and take a centered crop of spatial size $224 \times 224$. In contrast to Krizhevsky et al. (2012) we limit preprocessing to subtraction and divison of trainset mean and standard deviation and do not include local response normalization layers. We randomly augment training data with random horizontal flips. We set an initial learning rate of 0.1 and follow the learning rate schedule proposed in Krizhevsky et al. (2012) that drops the learning rate by a factor of 0.1 every 30 epochs and train for a total of 74 epochs.

The amount of epochs for the expansion of architectures is larger due to the re-initialization. For these architectures the mentioned amount of epochs corresponds to training during stable conditions, i.e. no further expansion. The procedure is thus equivalent to training the converged architecture from scratch.

## A.3  ARCHITECTURES

GFCNN (Goodfellow et al., 2013) Three convolution layer network with larger filters (followed by two fully-connected layers, but without "maxout". The exact sequence of operations is:

1. Convolution 1: $8 \times 8 \times 128$ with padding $= 4 \rightarrow$ batch-normalization $\rightarrow$ ReLU $\rightarrow$ max-pooling $4 \times 4$ with stride $= 2$.
2. Convolution 2: $8 \times 8 \times 198$ with padding $= 3 \rightarrow$ batch-normalization $\rightarrow$ ReLU $\rightarrow$ max-pooling $4 \times 4$ with stride $= 2$.
3. Convolution 3: $5 \times 5 \times 198$ with padding $= 3 \rightarrow$ batch-normalization $\rightarrow$ ReLU $\rightarrow$ max-pooling $2 \times 2$ with stride $= 2$.
4. Fully-connected 1: $4 \times 4 \times 198 \rightarrow 512 \rightarrow$ batch-normalization $\rightarrow$ ReLU.
5. Fully-connected 2: $512 \rightarrow$ classes.

Represents the family of rather shallow "deep" networks.

VGG (Simonyan & Zisserman, 2015) "VGG-A" (8 convolutions) and "VGG-E" (16 convolutions) networks. Both architectures include three fully-connected layers. We set the number of features in the MLP to 512 features per layer instead of 4096 because the last convolutional layer of these architecture already produces outputs of spatial size $1 \times 1$ (in contrast to $7 \times 7$ on ImageNet) on small datasets. Batch normalization is used before the activation functions. Examples of stacking convolutions that do not alter spatial dimensionality to create deeper architectures.

WRN (Zagoruyko & Komodakis, 2016) Wide Residual Network architecture: We use a depth of 28 convolutional layers (each block completely coupled, no bottlenecks) and a width-factor of 10 as reference. When we expand these networks this implies an inherent coupling of layer blocks due to dimensional consistency constraints with outputs from identity mappings.

Alexnet (Krizhevsky et al., 2012) We use the all convolutional variant where we replace the first fully-connected large $6 \times 6 \times 256 \rightarrow 4096$ layer with a convolution of corresponding spatial filter size and 256 filters and drop all further fully-connected layers. The rationale behind this decision is that previous experiments, our own pruning experiments and those of Hao et al. (2017); Han et al. (2015), indicate that original fully-connected layers are largely obsolete.

## A.4    Increasing a network's representational capacity beyond the reference when strong regularization is applied

As previously explained in the main-body in section 2 we generally differentiate between a deep neural network model's *representational* and *effective capacity*. Whereas the former only includes the layer and feature choices, the latter reflects the actual capability of a neural network to fit the data when including the choice of optimization and corresponding regularization. In practice this means that a network can manage to under-fit on the training data if those parameters aren't chosen appropriately, even if the network itself is comprised of an abundant amount of parameters.

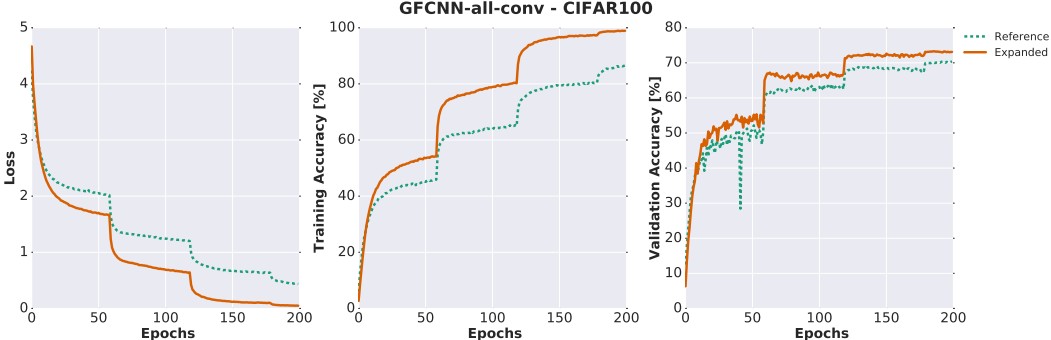

Figure 4: Left to right: Loss, train and validation curves for a GFCNN-all-conv trained on the CIFAR100 dataset. The dashed curve presents a reference architecture implementation whereas the solid lines correspond to the behavior of an expanded network (Note that we have omitted the non-stable part of the expansion and only show the 200 stable epochs). It can be observed that loss and training accuracy improve by a large margin for the expanded architecture, whereas validation accuracy benefits only slightly.

In figure 4 we show an example of such behavior on a GFCNN-all-conv architecture trained on the CIFAR100 dataset and what the implications are with respect to algorithm 1 and the expansion framework introduced in this work. The original architecture can be observed to severely under-fit the train-data which is largely due to the use of $L_2$ regularization, dropout and batch-normalization at the same time. Without lifting these constraints we can observe that our expanded architecture bridges this gap in loss and train-accuracy completely by increasing the width of the layers and thus allocating a lot more parameters (see 1 where we observe a $\approx 5\times$ increase in parameters). We can further observe that the validation accuracy benefits only slightly from this increase (here $\approx 3\%$). In other words, our expansion algorithm tries to counter the under-fitting due to the heavy

regularization in order to fit the training set, which results in some over-fitting in return. However we would like to make the following two remarks of why we believe this could be desirable: 1.) A model that severely under-fits (whether due to dropout, loss function regularization terms, batch normalization etc.) will unarguably be incapable of generalizing very well. 2.) The model seems to nevertheless benefit from the increased capacity and strong regularization, even if not to full extend with respect to generalization.

### A.5 A BRIEF DISCUSSION OF FEATURE RE-INITIALIZATION

We have empirically observed promising results even without the re-initialization included in algorithm 1. However, we have currently chosen not to report any experiments as we believe deeper analysis of factors such as as stability and feature initialization is required at this stage to avoid misleading results. Some of the concrete open questions that we believe need to be analyzed more thoroughly for such an approach to be employed in practice are:

#### INITIALIZATION OF NEW FEATURES

Initialization of new features during training: for a used initialization scheme, e.g. Xavier (Glorot & Bengio, 2010) or Kaiming He et al. (2015) initialization, is such a scheme employed throughout the entire process of network expansion? Adopting such practice would initialize newly added features differently from previously initialized features due to varying Fan-in and Fan-out dimensionality. Depending on the precise nature of the initialization scheme each subsequent feature would be scaled to decreasing or increasing magnitude, which could lead to undesired behavior such as some features not being used or old features becoming obsolete.
One potential solution could be to make sure that newly added features are aligned in a sense that they follow the distribution of already learned features. Explicitly, when drawing the first features from e.g. a normal distribution with mean 0 and some std, new features could stem from a normal distribution with mean and std of already learned features.

#### STABILITY OF EXPANSION

Whenever a new filter without re-initialization is introduced, although the currently learned knowledge base is not modified, a perturbation to the classifier is introduced. Let us assume the extreme case where the initial amount of training epochs was 200 and we make an incremental addition of 1 or 2 features in epoch 199. Although a classifier should be able to rapidly recover and get back to high accuracy by fine-tuning, this fine-tuning is also needed to make sure the training does not end in perturbed state. In addition the magnitude of the perturbation depends on the size of the already existing feature base, leading to another important factor that should be regarded.

### A.6 AUTOMATICALLY EXPANDED ARCHITECTURE TOPOLOGIES

In addition to figure 3 we show mean evolved topologies including standard deviation for all architectures and datasets reported in table 1 and 2. In figure 5 and 6 all shallow and VGG-A architectures and their respective all-convolutional variants are shown. Figure 7 shows the constructed wide residual 28 layer network architectures where blocks of layers are coupled due to the identity mappings. Figure 8 shows the two expanded Alexnet architectures as trained on ImageNet.
As explained in the main section we see that all evolved architectures feature topologies with large dimensionality in early to intermediate layers instead of in the highest layers of the architecture as usually present in conventional CNN design. For architectures where pooling has been replaced with larger stride convolutions we also observe that dimensionality of layers with sub-sampling changes independently of the prior and following convolutional layers suggesting that highly-complex pooling operations are learned. This an extension to the proposed all-convolutional variant of Springenberg et al. (2015), where introduced additional convolutional layers were constrained to match the dimensionality of the previously present pooling operations.

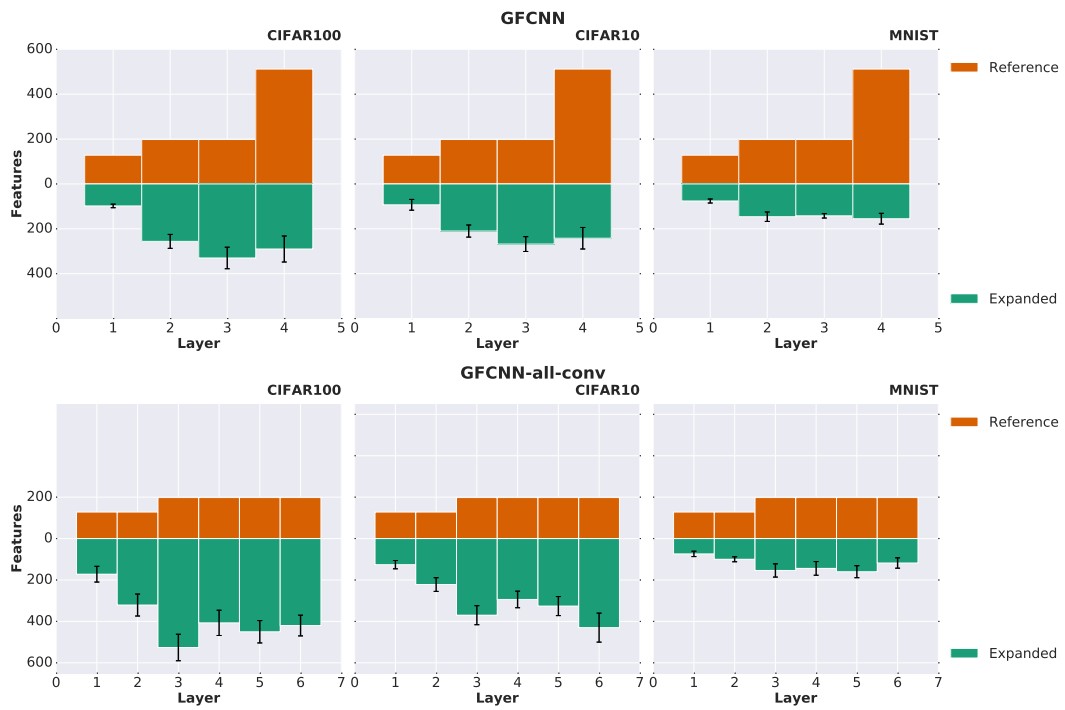

Figure 5: Mean and standard deviation of topologies as evolved from the expansion algorithm for the shallow networks run five times on MNIST, CIFAR10 and CIFAR100 datasets respectively.

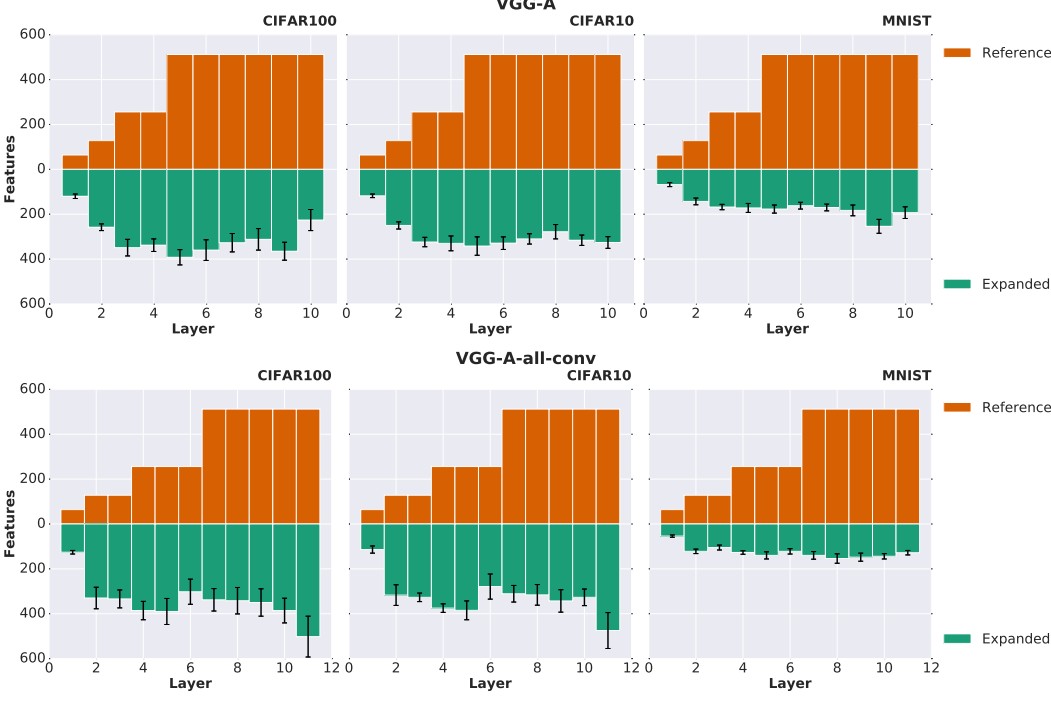

Figure 6: Mean and standard deviation of topologies as evolved from the expansion algorithm for the VGG-A style networks run five times on MNIST, CIFAR10 and CIFAR100 datasets respectively.

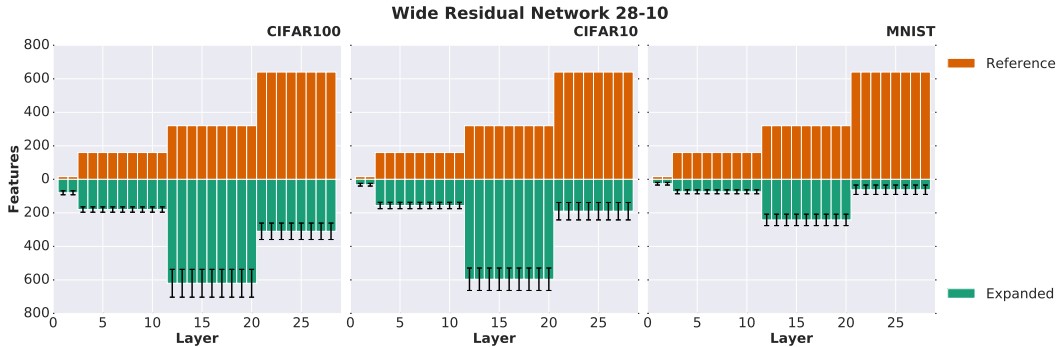

Figure 7: Mean and standard deviation of topologies as evolved from the expansion algorithm for the WRN-28 networks run five times on MNIST, CIFAR10 and CIFAR100 datasets respectively. Note that the CIFAR100 architecture was limited in expansion by hardware.

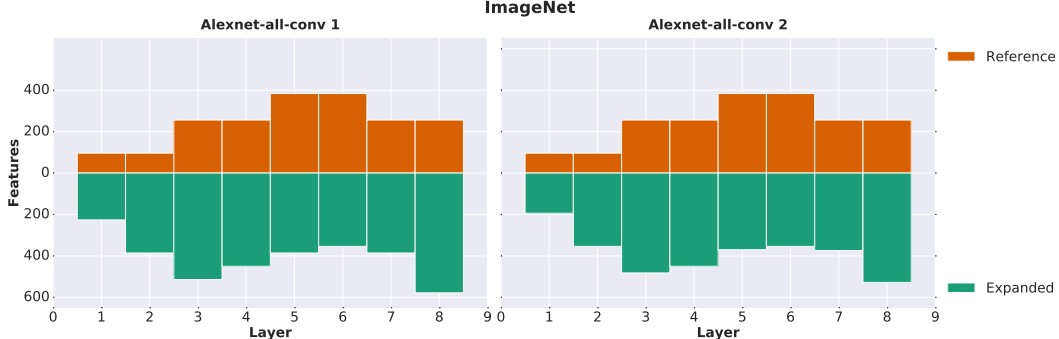

Figure 8: Architecture topologies for the two all-convolutional Alexnets of table 2 as evolved from the expansion algorithm on ImageNet.

