# OpenReview forum: "Building effective deep neural networks one feature at a time"
_ICLR.cc/2018/Conference — Reject_

### Official Review · AnonReviewer1 · 2017-11-27
**Greedy network feature depth optimization**

**Rating:** 4
**Confidence:** 5

**Review:**

The authors propose an approach to dynamically adjust the feature map depth of a fully convolutional neural network. The work formulates a measure of self-resemblance, to determine when to stop increasing the feature dimensionality at each convolutional layer. The experimental section evaluates this method on MNIST, CIFAR-10/100 and a limited evaluation of ImageNet. Generally, I am a very big proponent of structure learning in neural networks. In particular, we have seen a tremendous boost in performance in going from feature engineering to feature learning, and thus can expect similar effects while learning architectures rather than manually designing them. One important work in this area is "Self-informed neural network structure learning" by Farley et al. that is missing from the citations.
However, this work falls short of its promises.

1. The title is misleading. There really isn't much discussion about the architecture of networks, but rather the dimensionality of the feature maps. These are very different concepts.
2. Novelty of this work is also limited, as the authors acknowledge, that much of the motivation is borrowed from Hao et al., while only the expansion mechanism is now normalized to avoid rescaling issues and threshold tuning.
3. The general approach lacks global context. All decisions about individual feature depths are made locally both temporally and spatially. In particular, expanding the feature depth at layer f at time t, may have non trivial effect on layer f-1 at time t + 1. In other words, there must be some global state-space manifold to help make decisions globally. This resembles classical dynamic programming paradigms. Local decisions aren't always globally optimal.
4. Rather than making decision on per layer basis at each iteration, one should wait for the model to converge, and then determine what is useful and what is not.
5. Finally, the results are NOT promising. In table 1, although the final error has reduced in most cases, it comes at the expense of increases capacity, in extreme cases as much as ~5x, and always at the increased training time, in the extreme case ~14x, An omitted citation of "Going deeper with Convolution" is an example, where a much smaller footprint leads to a higher performance, further underlying the importance of a smaller footprint network as stated in the abstract.

---

> ### Public Comment · (anonymous) · 2017-12-14
> **AnonReviewer1 Rebuttal**
>
> Thank you for taking the time to read our work and write this review. We share the view that automated neural network design holds large promise. Concerning the five points we are a little dismayed by the statements.
>
> 1.) In our opinion the word “architecture” doesn’t have a rigid definition and can span a variety of concepts. We have chosen the title because we investigate different neural networks and notice common patterns in formation of feature space dimensionality. We think that the abstract makes the scope of the paper quite clear. If it is allowed, we can imagine omitting the word “architecture” in the title. We believe this should clear the confusion.
>
> 2.) >“lack of novelty” and the statement that our work is largely “borrowed”.
>  While Hao et al. provided inspiration, there are several crucial differences to “Pruning Filters for Efficient Convnets”:
>
> -Hao et al ONLY talks about pruning already trained NNs. Our metric follows in spirit by observing entire filters instead of individual weight values. But while Hao et al base pruning on filter magnitudes, we look at the evolution over time in a normalized fashion. Due to this change we can move to a BOTTOM-UP expansion approach instead of pruning. This is fundamentally different from any pruning paper. We do NOT present this work as a technique for pruning at all.
>
> -“ while only the expansion mechanism is now normalized to avoid re-scaling issues and threshold tuning.”
> The expansion mechanism itself is novel and only works due to the added idea of normalization. To the best of our knowledge this has not been proposed in previous works. Works in the spirit of Hao et al. take top-down approaches where it is always required to train a neural network to full convergence first. A network’s feature dimensionality had to first be picked through large scale empirical experimentation and human intuition before pruning. In contrast, our work incrementally adds capacity starting from just one feature per layer.
>
> -We empirically observe alternate feature composition in comparison with the common rule of thumb for NN design of adding features towards deeper layers. We speculate that this could play a role in future NN design.
>
> 3.) >“expanding the feature depth at layer f at time t, may have non trivial effect on layer f-1 at time t + 1”.
> We agree that change in number of features in any layer has non trivial effect on the other layers. This is the primary reason why we re-initialize features every time a feature is added to avoid the introduction of non-trivial perturbations.
>
> >“Local decisions aren't always globally optimal.”
> Temporal evolution of weights is very much dependent on the minimization of the global cost. In a layer that already has more than required number of features some of the features will not receive any or minor update from the SGD step. Based on our metric no further addition of features will be required. Since it is a greedy approach, we cannot guarantee global optimality, but under given regularization constraints, our approach seems to find a good solution without loss or even improvement of generalization.
>
> 4.) In our opinion pruning is a good approach for parameter reduction in models. In the context of moving towards automation of network building, one has to still decide on what network size to train to convergence before pruning. Identification of suitable feature dimensionalities for unknown datasets by itself is a difficult and demanding task. To an extent, our expansion approach aims to overcome this limitation as adequate feature dimensionalities are approached in a bottom-up fashion.
>
> 5.) > Model complexity: Our expansion mechanism operates on the basis of temporal evolution of weights which depends on the ability of the model to push for zero training error under regularization constraints. In some cases, e.g. 5x increase in parameters, the corresponding original models underfit on the training data with this particular hyperparameter and regularizer configuration. It is understandable that our approach adds parameters to build a model which adequately fits the training data. Arguably a model that underfits on training data (whether due to dropout, loss function regularization terms, batch normalization etc.) will not be able to generalize well either. We will add an appropriate example with loss, train and validation curves to improve the readers understanding.
>
> > Training time: We believe it is unfair to compare the time of our approach against original models. One should recognize that authors of the original models arrive at those architectures after rigorous experimental validation of many feature configurations which all together takes lot more time. Our approach on the other hand, given the depth of network, starts with one feature per layer and automatically chooses suitable feature dimensionality in one go.

---

> ### Public Comment · (anonymous) · 2017-12-14
> **Reference to Farley et al. "Self-informed neural network structure learning"**
>
> We welcome the additional reference to Farley et al’s work. We went through it carefully and believe that the work contains some great ideas. We would also like to point out that the scope of “Self-informed neural network structure learning” is different and in fact orthogonal to the work presented here.
>
> Our work is complementary in a sense that it could be used as a precursor. Farley et al. show how to adapt/transfer already well-performing trained networks by doing capacity increases (e.g. with a large ImageNet trained GoogLeNet), whereas our work tackles the challenge of coming up with suitable capacity and feature spaces of such a network in the first place. In our understanding, Farley et al’s work does not seem to focus on the question of whether the underlying trained neural network’s capacity is appropriate in the first place and relies on this factor as given.
> In this sense, our proposed method is valuable in construction of the initial feature space (from very small to larger more adequate) capacity on a task, and the method suggested by Farley could offer incremental capacity addition on top of the converged architecture when moving to novel data. We had thus initially not cited this work, but will include a reference to Farley et al in the related work section as a valuable orthogonal idea.

---

### Official Review · AnonReviewer3 · 2017-11-28
**simple idea that is shown to work well in practice, preliminary ImageNet results demonstrate scalability**

**Rating:** 8
**Confidence:** 4

**Review:**

This paper introduces a simple correlation-based metric to measure whether filters in neural networks are being used effectively, as a proxy for effective capacity. The authors then introduce a greedy algorithm that expands the different layers in a neural network until the metric indicates that additional features will end up not being used effectively.

The application of this algorithm is shown to lead to architectures that differ substantially from hand-designed models with the same number of layers: most of the parameters end up in intermediate layers, with fewer parameters in earlier and later layers. This indicates that common heuristics to divide capacity over the layers of a network are suboptimal, as they tend to put most parameters in later layers. It's also nice that simpler tasks yield smaller models (e.g. MNIST vs. CIFAR in figure 3).

The experimental section is comprehensive and the results are convincing. I especially appreciate the detailed analysis of the results (figure 3 is great). Although most experiments were conducted on the classic benchmark datasets of MNIST, CIFAR-10 and CIFAR-100, the paper also includes some promising preliminary results on ImageNet, which nicely demonstrates that the technique scales to more practical problems as well. That said, it would be nice to demonstrate that the algorithm also works for other tasks than image classification.

I also like the alternative perspective compared to pruning approaches, which most research seems to have been focused on in the past. The observation that the cross-correlation of a weight vector with its initial values is a good measure for effective filter use seems obvious in retrospect, but hindsight is 20/20 and the fact is that apparently this hasn't been tried before. It is definitely surprising that a simple method like this ends up working this well.

The fact that all parameters are reinitialised whenever any layer width changes seems odd at first, but I think it is sufficiently justified. It would be nice to see some comparison experiments as well though, as the intuitive thing to do would be to just keep the existing weights as they are.

Other remarks:

Formula (2) seems needlessly complicated because of all the additional indices. Maybe removing some of those would make things easier to parse. It would also help to mention that it is basically just a normalised cross-correlation. This is mentioned two paragraphs down, but should probably be mentioned right before the formula is given instead.

page 6, section 3.1: "it requires convergent training of a huge architecture with lots of regularization before complexity can be introduced", I guess this should be "reduced" instead of "introduced".

---

> ### Public Comment · (anonymous) · 2017-12-13
> **Comments for AnonReviewer3**
>
> Thank you very much for taking the time to write this review.
>
> >“That said, it would be nice to demonstrate that the algorithm also works for other tasks than image classification.“
>
> Thank you very much for the pointer. We are planning on and will make sure to add experiments on other data types in the future.
>
> > “The fact that all parameters are re-initialised whenever any layer width changes seems odd at first, but I think it is sufficiently justified. It would be nice to see some comparison experiments as well though, as the intuitive thing to do would be to just keep the existing weights as they are.”
>
> We think that adding these experiments should only be hinted at and largely be postponed to a later version together with a more rigorous analysis. The reasoning here is similar to what we have stated in the outlook as we believe that it is necessary to do a more profound analysis of initialization techniques and the accompanied effect on convergence behavior. Given that the open questions here are of untrivial nature, we were hesitant to simply include some experiments here.
> We are going to outline some of the concrete questions about initialization (re-initialization) more thoroughly in the future work section to give the reader a better understanding of the challenges and possibilities when moving away from re-initialization.
>
> >“Formula (2) seems needlessly complicated because of all the additional indices”
>
> Thank you for the suggestion, we do agree that there can be less amount of indices. We chose to explicitly write down all the indices to avoid any ambiguity. We agree that we can simplify the spatial indices and only make the incoming and outcoming feature dimensionality explicit. We will update this. We will also make sure to mention that our equation is basically normalized cross-correlation right next to the formula to improve understanding as well.

---

### Official Review · AnonReviewer2 · 2017-11-30
**Not sure about the novelty / contribution of the paper.**

**Rating:** 5
**Confidence:** 4

**Review:**

This paper aims to address the deep learning architecture search problem via incremental addition and removal of channels in intermediate layers of the network. Experiments are carried out on small-scale datasets such as MNIST and CIFAR, as well as an exploratory run on ImageNet (AlexNet).

Overall, I find the approach proposed in the paper interesting but a little bit thin in content. Essentially, one increases or decreases the number of features based on equation 2. It would be much valuable to see ablation studies to show the effectiveness of such criterion: for example, simple cases one can think of is to model (1) a data distribution of known rank, (2) simple MLP/CNN models to show the cross-layer relationships (e.g. sudden increase and decrease of the number of channels across layers will be penalized by c^l_{f^{l+1}, t}), etc.

The experimentation section uses small scale datasets and as a result, it is relatively unclear how the proposed approach will perform on real-world applications. One apparent shortcoming of such approach is that training takes much longer time, and the algorithm is not easily made parallel (the sgd steps limit the level of parallelization that can be carried out). As a result, I am not sure about the applicability of the proposed approach.

---

> ### Public Comment · (anonymous) · 2017-12-13
> **AnonReviewer2 Rebuttal**
>
> Thank you very much for the review and suggestions on how to improve our work. We would like to request the reviewer for some further clarification which will help us in the improvements.
>
> > “It would be much valuable to see ablation studies to show the effectiveness of such criterion: for example, simple cases one can think of is to model (1) a data distribution of known rank, (2) simple MLP/CNN models to show the cross-layer relationships (e.g. sudden increase and decrease of the number of channels across layers will be penalized by c^l_{f^{l+1}, t}), etc.”
>
> We propose our approach as a greedy expansion method to construct a network’s feature space such that it can fit underlying data under some regularization constraints. Given a fixed amount of layers, we start with one feature per layer and grow the capacity until the network is capable of adequately fitting the training data.
> With respect to comment (1)(“a data distribution of known rank”), we agree that suggested analysis will be very important and further theoretical analysis will be valuable and is necessary.
> We also believe that comment (2)(“show cross-layer relationships”) addressing understanding of cross layer relationships in neural networks and analysis of multi-layer non-linear networks’ feature spaces can provide further insights.
>
> Following the reviewer’s suggestion, we can imagine conducting some toy dataset experiment of the following type:
> Take data distributions of increasing rank. Monitor and analyse the relationship to the capacity that our expansion algorithm allocates depending on rank.
> However it is unclear to us how such an analysis will provide more rigorous insights into the cross-layer relationships or how increase of distribution rank maps to a multi-layer non-linear neural network (even if we talk about a multi hidden layer MLP), particularly under regularization and SGD sampling. Unless we conduct such an experiment for a very shallow, linear MLP, to the best of our knowledge this will result in purely empirical insights on whether the capacity allocated by our algorithm scales (in a similar fashion as observed when moving from MNIST to CIFAR10 to CIFAR100). We would be grateful if the reviewer could further clarify his suggestion.
>
> It is our view that the novel contributions in this work are 1) the expansion framework for network building itself, 2) conducted experiments and  3) the idea to use a normalized cross-correlation metric.
>
> >“The experimentation section uses small scale datasets “
>
> We included a few initial experiments on the large scale ImageNet dataset in the initial version submitted for the review and hope to add more in the future. We would kindly ask the reviewer to consider that running experiments on ImageNet using very large architectures like ResNets and its corresponding hardware demands is a resource challenge for many.
>
> >“One apparent shortcoming of such approach is that training takes much longer time.”
>
> It is true that our approach takes longer compared to the pure training time of corresponding original models. However, we believe that one should also take into account all the time spent by the authors of original models on validation experiments (grid-search, random-search etc.) used to find those models. In general in this paper we present that our method works consistently for the datasets tested so far.
> If we imagine an encounter with a new (vision) dataset of unknown origin and the task to find a suitable (convolutional) neural network. Our method can provide substantial benefits in exploring architecture options by not having to choose feature dimensionality by hand and e.g. concentrating on amount of layers instead. The alternatives are all very time-consuming if the user doesn’t already have a very good prior on task complexity (like on the common benchmark datasets) and often includes training of networks that initially completely over- or underfit before determining suitable upper or lower bounds on model complexity.
>
> >“the algorithm is not easily made parallel (the sgd steps limit the level of parallelization that can be carried out)”
>
> To summarize our expansion approach, we start with a network consisting of one feature per layer and begin training. Based on the temporal evolution metric proposed, features are added at each layer and training is re-initialized, very much like training a new network with increased features per layer. This process repeats until no more features are added at each layer and the final network is trained to convergence.
> If we now compare this to traditional SGD (and its variants) with a fixed model that means that we do not interfere with the optimization, other than making the decision of whether to expand after a SGD step is taken.
>
> We would greatly appreciate if you could further elaborate on this point as it isn’t clear to us how SGD steps are limiting the level of parallelization in our approach in contrast to “conventional” SGD.

---

### Public Comment · ~W._James_Murdoch1 · 2017-10-20
**Related work**

I think this paper from ICLR 2017 may be relevant to your work, and is probably worth adding to your related work section.

Best of luck

https://www.cs.cmu.edu/~jgc/publication/Nonparametric%20Neural%20Networks.pdf

---

> ### Author Response · Authors · 2017-11-06
> **Related work**
>
> Thank you for the pointer to the ICLR 2017 paper. We were presently unaware of this paper, but after taking a brief look identified it as a relevant reference.
>
> We will go through it more thoroughly and then add it where appropriate in the related work section.
>
> Best,

---

### Author Response · Authors · 2018-01-05
**Revision including reviewer feedback**

We would like to briefly remark that there seems to have been some difficulty in posting the rebuttal as an "official comment" at the time. To clarify, the "anonymous" comments marked with "rebuttal" and the "Comments for AnonReviewer3" have been posted by this paper's authors and should be regarded as official comments.

We again thank the reviewers for their efforts and have uploaded a revised document improving upon suggested aspects wherever possible. To give a short summary we have:

* included 2 suggested valuable references into related work
* made a minor modification to the title by omitting the word "architectures" and instead simply writing "neural networks" as reviewer 1 has kindly noted that the word and concept of architectures seems to have different interpretations in the community and thus could be misleading in the title of our work.
* added an additional appendix section discussing increase of networks' capacities beyond the reference (addressing reviewer 1). We provide an example with loss, training and validation curves to show the non-triviality of effective capacity when regularizers are present.
* made minor modifications to the main body to further underline the novelty to the reader and avoid miss-conceptions about concepts being borrowed by "Hao et. al." or other pruning papers that are not in the scope of the expansion framework presented in this work. (addressing reviewer 1)
* simplified equation 2 with respect to the explicit indices of the norm and the spatial dimensions. We have furthermore made corresponding changes to the description of the equation to portray the cross-correlation concept earlier. This should improve readability and understanding of the equation. (addressing reviewer 3)
* added a section in the appendix addressing the possibility and open questions of applying our proposed framework without the need for re-initialization. The section should further clarify why we have decided to not include a demo of such an experiment as we believe it would lead to potentially misleading results and interpretation. (addressing reviewer 3)
* made minor modifications to wording and corrected some few typos.
* rephrased a short part about the computational perspective of our approach to emphasize the approach's modularity and potential for parallelization with no limitations known to us beyond regular SGD optimization (addressing reviewer 2)

Unfortunately we have not been able to include the request made by reviewer 2: "data distribution of known rank and simple models to show cross-layer relationships". We have thought long and hard about this statement and could not come to a conclusion of how to conduct such an experiment in a convincing manner. We believe that such experiments about cross-layer relationships are absolutely desirable, but still an open-challenge for deep learning in general and thus not immediate to our contribution. We have requested some clarification about the nature of such experiments and did not yet receive further explanation. Independent of the decision of acceptance of our work we would be extremely grateful if the reviewer could extend and clarify the review so that we can draw more value from it and include it in future work and improvements.

As a last remark we would like to again point out our concern with the very harsh lack of novelty statement made by reviewer 1. The reviewer seems to believe our mechanism is "borrowed" from Hao et al's paper, which is concerned _only_ with pruning of already _trained_ networks, and voices correspondingly harsh feedback about the value (or lack there-of) of our bottom-up expansion approach. We are particularly concerned with the one-sided nature of statements such as "one should wait for the model to converge, and then determine what is useful and what is not.". Independent of whether such a statement turns out to be true, we strongly believe that exploration of alternatives (one presented here) to always training networks to full convergence before making modifications is crucial and provides necessary insights beyond "pushing benchmark numbers".

---

### Decision · Program_Chairs · 2018-01-29
**ICLR 2018 Conference Acceptance Decision**

**Decision:**

Reject

**Comment:**

Regarding clarity, while the paper definitely needs work if it is to be resubmitted to an ML venue, different revisions would be appropriate for a physics audience. And given the above comment, any suggested changes are likely to be superfluous.